# Incremental Hierarchical Reinforcement Learning with Multitask LMDPs

## Abstract

Exploration is a well known challenge in Reinforcement Learning. One principled way of overcoming this challenge is to find a hierarchical abstraction of the base problem and explore at these higher levels, rather than in the space of primitives. However, discovering a deep abstraction autonomously remains a largely unsolved problem, with practitioners typically hand-crafting these hierarchical control architectures. Recent work with multitask linear Markov decision processes, allows for the autonomous discovery of deep hierarchical abstractions, but operates exclusively in the offline setting. By extending this work, we develop an agent that is capable of incrementally growing a hierarchical representation, and using its experience to date to improve exploration.

## 1 Introduction

The exploration-exploitation trade-off is much studied in the Reinforcement Learning (RL) literature. A central question regards how to explore efficiently in large state spaces. In this direction, a number of classical exploration ideas have emerged for problems with enumerable state spaces which could be represented in tabular format Roger (1999). In fact, approximately optimal exploration is achievable in these settings Kolter & Ng (2009). When the state space is large and neural networks are used to approximate the value function, an additional suite of techniques exist to promote exploration Stadie et al. (2015); Tang et al. (2016); Plappert et al. (2017).

An always good-in-principle idea is to first construct an abstract representation of the problem, and explore at this higher level rather than in the space of primitives. This might correspond to a person exploring a house by choosing to explore at the level of rooms, rather than exploring exclusively inch-by-inch. Intuitively, this suggests that exploring a large state space may be best conducted at different levels of granularity, corresponding to different layers of abstraction. The challenge here is that finding suitable abstractions autonomously is, in general, largely an unsolved problem, although some progress has been made in this direction Vigorito & Barto (2010); Bacon & Precup (2015). Additionally, while further layers of abstraction would enhance the agent's ability to explore at these various scales, many architectures in the hierarchical reinforcement learning literature focus on just two layers of control, e.g., the options framework Sutton & Singh (2015); Machado et al. (2018).

Recent work using the linearly-solvable Markov decision process (LMDP) framework provides a mechanism for autonomously learning deeper control hierarchies Saxe et al. (2017). However, this method operates in the off-line setting. By incorporating ideas from incremental learning on static classification tasks, we develop an agent that is able to both build, and utilize, a hierarchical architecture fully online. Our agent operates by periodically increasing the capacity of its hierarchical representation, and allowing subsequent experience to fine-tune the new weights. In so much as an analogy to the well-known options framework is helpful, the new approach can be thought of as periodically adding a randomly initialized options policy, and allowing subsequent experience to adjust all of the available options in light of the new-found capacity.

We demonstrate that learning the hierarchical representation online is not only possible, but can actually aid exploration when coupled with a simple count-based exploration boost.

## 2 PRELIMINARIES

The LMDP framework Todorov (2006); Kappen (2005) considers a special class of MDPs for which the Bellman optimality condition becomes linear in the exponentiated cost-to-go. In this formalism, the MDP is defined as a three-tuple $L = \langle S, P_{\bar{\pi}}, R \rangle$, where $S = [1, \ldots, N]$ is a set of states, and $P_{\bar{\pi}} : S \times S \to [0, 1]$ is the so-called *passive* transition probability between states. This transition probability is defined as the resultant transition probability under some reference policy $\bar{\pi}$. And $R : S \to \mathbb{R}$ is the expected instantaneous reward. The reference policy $\bar{\pi}$ is typically taken to be the uniformly random policy, corresponding to the entropy-regularized RL problem Schulman et al. (2017); Haarnoja et al. (2017). Intuitively, the passive dynamics can be thought of as the diffuse process followed by the agent in the absence of a specified control.

In this setting, the action taken by the agent is a full distribution over next states $a(\cdot|s) \in \mathbf{R}^N$, which acts to shift the passive transition probability by redistributing probability mass; making preferred transitions more likely, and disfavoured transitions less likely. A control cost is associated with this choice: actions corresponding to distributions over next states that are very different from the passive transition probability distribution are expensive, while those that are similar are cheap. In this way the problem is said to be regularized by the reference policy. Intuitively, this can be thought of as specifying an intrinsic preference for energy-efficient actions, by regularizing the controlled transition probabilities towards the reference policy. The LMDP has rewards $R_i(s)$ for each interior state, and $R_b(s)$ for each boundary state in the finite exit formulation. The LMDP can be solved by finding the so-called *desirability* function $z(s) = e^{V(s)}$, specified in terms of the state value function. The optimal policy can then be computed in closed form as:

$$a^*(s'|s) = \frac{P_{\bar{\pi}}(s'|s)z(s')}{\mathcal{G}[z](s)}, \tag{1}$$

where the normalizing constant $\mathcal{G}[z](s) = \sum_{s'} P_{\bar{\pi}}(s'|s)z(s')$ Todorov (2009). Intuitively, the hard maximization of standard MDP has been replaced by a soft maximization $\log(\sum \exp(\cdot))$, and the continuous action space enables closed form computation of the optimal policy.

While in principle any reference policy could be used (perhaps coming from an expert demonstration), entropy regularization is a common choice, and has been shown to recover more adaptive, multi-modal policies in large domains Haarnoja et al. (2017). As an illustrative example, consider a robot whose task it is to navigate around some object. Suppose that two equally valid solutions exist, corresponding to a trajectory around the obstacle to the left, and another to the right. Where a standard DQN-style agent would collapse on one path or the other, in the entropy regularized problem the optimal solution is a stochastic policy in which the agent chooses which path to take at random. Suppose that, after training, the path to the left was blocked-off by a new obstacle. While the DQN-style agent would struggle to recover from its firm commitment to the left path, the entropy-regularized agent is able to adapt easily.

It is worth noting that unlike the standard MDP formulation, the optimal policies uncovered in the LMDP formulation are stochastic. These have equivalent interpretations as being stochastic policies over deterministic actions, or as deterministic policies over stochastic actions. Explicitly, although the LMDP formalism transforms away the notion of discrete actions, the optimal solutions to LMDP problems may always be realized in terms of stochastic policies over the discrete actions of the more familiar MDP formulation.

### 2.1 THE MULTITASK LMDP

The linearity of the Bellman optimality condition implies a natural form of policy composition. Suppose that we solve two LMDPs $L^1 = \langle S, P_{\bar{\pi}}, [R_i, R_b^1] \rangle$ and $L^2 = \langle S, P_{\bar{\pi}}, [R_i, R_b^2] \rangle$, which operate in the same state and action space, but differ in their instantaneous boundary rewards, and obtain the corresponding optimal policies $z^1$ and $z^2$. If we then encounter a third LMDP $L^3 = \langle S, P_{\bar{\pi}}, [R_i, R_b^3] \rangle$ such that if $r_b^3 = \alpha r_b^1 + \beta r_b^2$, then the corresponding optimal policy is immediately realizable as $z^3 = \alpha z^1 + \beta z^2$ Todorov (2009). Importantly, no further learning need take place.

This property was exploited in Saxe et al. (2017), to develop a powerful multitasking framework. In this setting, an agent faces a number of tasks which have the same state and action spaces, but differ

in their boundary reward functions. The multitask LMDP (MLMDP) operates by learning a set of $N_t$ tasks each defined as an LMDP $L_t = \langle S, P, R = [r_i, r_b^t] \rangle$, $t = 1, \cdots, N_t$. This set of LMDPs represents an ensemble of tasks with different ultimate goals. These tasks may then be collected into a task basis (or task library). By utilizing this library of pre-learned policies, when the agent encounters a new task, it is able to immediately obtain the optimal policy by representing the policy for the new task as a weighted blend of policies from the task library. If the new task is not exactly realizable as a linear combination of tasks from the library, significant jump-start performance may nevertheless be obtained by initializing the new task as an approximate blend of library tasks Saxe et al. (2017). A particular instantiation of a set of tasks is referred to as being a task 'module'.

## 2.2 THE HIERARCHICAL MLMDP

Hierarchical RL holds the promise of dramatically simplifying the learning problem by abstracting away the finer details of the base problem, and instead solving a consistent, but simpler, higher layer problem. By repeatedly applying this abstraction procedure, an agent may in principle overcome the curse of dimensionality which typically occurs as the size of the state and action spaces increase. Of course, this simply replaces one difficulty with another: instead of directly solving the original problem, we need to solve the new problem of how to uncover a suitable abstraction in the first place. Typically this abstraction is hand-crafted by a designer, however a number of techniques have been proposed for autonomously uncovering useful abstractions under the banner of "subtask/options discovery" Konidaris (2016); Stolle & Precup (2002); Bonarini et al. (2006).

These discovery procedures are themselves limited in that they uncover only a single layer of abstraction. It is often not at all clear how these procedure might be recursed to uncover deep representations akin to 'options-over-options' (but see Vigorito & Barto (2010)). Furthermore, abstractions that involve temporally extended actions often suffer from the fact that they artificially inflate the action space. This results in an exponential increase in the number of parameters to be learned, and is often crippling. However, by leveraging the compositionality of the MLMDP, a simple recursive scheme was described in Saxe et al. (2017) for constructing arbitrarily deep hierarchical abstractions. This is achieved by stacking the multitask module discussed above (Saxe et al., 2017). The stacking is realized by iteratively constructing higher order MLMDPs in which higher levels select the instantaneous reward structure that defines the current task for lower levels in a feudal-like architecture Dayan & Hinton (1993); Vezhnevets et al. (2017).

This recursive procedure is carried out by firstly augmenting the layer $l$ state space $\tilde{S}^l = S^l \cup S_t^l$ with a set of $N_t$ terminal boundary states $S_t^l$ called *subtask* states. Transitioning into a subtask state corresponds to a decision by the layer $l$ MLMDP to access the next level of the hierarchy. The transitions into the subtask states are governed by a new $N_t^l$-by-$N_i^l$ passive dynamics matrix $P_t^l$. In the augmented MLMDP, the full passive dynamics then become $\tilde{P}^l = [P_i^l; P_b^l; P_t^l]$, corresponding to transitions to interior states, boundary states, and subtask states respectively. The higher layer MLMDP is then defined by specifying transitions dynamics, $[P_i^{l+1}; P_b^{l+1}]$ and reward function $[r_i^{l+1}]$, such that the higher layer MLMDP is consistent with the lower layer MLMDP (Saxe et al., 2017), and solving the easier higher layer problem ultimately solves the base problem too. An example of this hierarchical scheme is shown in Fig.(1). Notice that the new subtask states form a separate *bona fide* LMDP $L^{l+1} = \langle S_t, [P_i^{l+1}, P_b^{l+1}], [R_i^{l+1}, R_b] \rangle$. This forms the basis for subsequent recursion.

Solving the higher layer MLMDP will yield an optimal action $a^{l+1}(\cdot|s)$. This action shifts the passive transition dynamics making some transitions more likely, indicating that they are desirable for the current task, and others less likely, indicating that they should be avoided for the current task. The instantaneous rewards for the lower layer are then set to be proportional to the difference between the controlled and passive dynamic, $r_t^l \propto a_i^{l+1}(\cdot|s) - p_i^{l+1}(\cdot|s)$.

In order to stack these modules, the subtask transition matrix $P_t^l$ must be defined at each layer. It was shown in Earle et al. (2018) that a suitable $P_t^l$ could be recovered autonomously by finding a low-rank approximation to the full task basis $Z^l \mathbf{R}^{N \times N}$ at each layer. Intuitively, this exploits the fact that there is often significant additional structure in the policies that make up the task basis. By way of example, suppose that we are in a 4-rooms domain, and the task basis is comprised of a policy to get to each individual state. A low-rank approximation of this basis should uncover the fact that the polices are grouped by the room contains their goal state. Since the task basis $Z^l$ is the

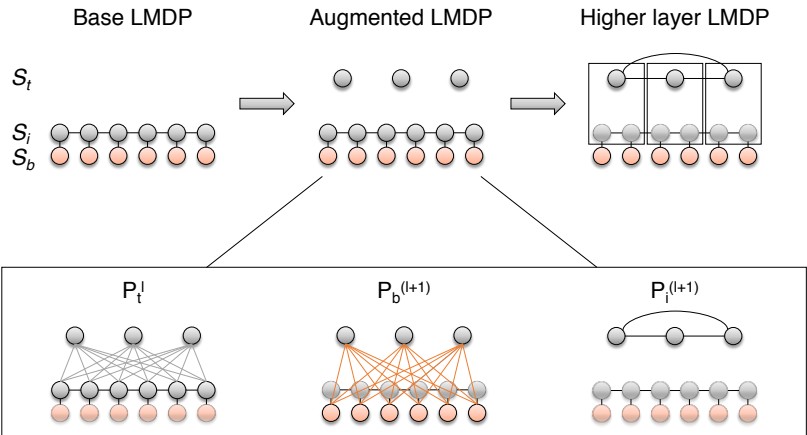

Figure 1: **Recursively constructing higher layer MLMDPs**. The base LMDP (LEFT) is specified as a set of interior and boundary states $S_i, S_b$, along with their corresponding instantaneous rewards $R_i, R_b$, and state transition dynamics $P_i, P_b$. The recursion is realized by first augmenting the base LMDP with an addition set of subtask states $S_t$ (CENTER). In order to successfully integrate these new states, the corresponding reward function $R_t$ and transition dynamics $[P_t, P_b^{l+1}, P_i^{l+1}]$ are suitably specified. The resulting construction is itself an LMDP, forming the basis for recursion (RIGHT).

exponentiated cost-to-go for the composite policies, and are therefore strictly non-negative, Earle et al. (2018) propose to utilize non-negative matrix factorization to uncover the low-rank approximation as $Z^l \approx D^l W^l$, and set $P_t^l = \alpha D^l$. This allowing for the fully autonomous discovery of deep control hierarchies. The method proceeds recursively by first constructing a task basis $\hat{Z}^l$, then finding a low rank approximation of that basis and assigning $P_t^l = \alpha D^l$, and finally computing the transition dynamics $P^{l+1}$ and reward structure $R^{l+1}$ for the abstracted MLMDP.

Howver, a key limitation of the method presented in Earle et al. (2018) is that the computation of the hierarchy happens in an off-line setting. While the method is therefore applicable to probabilistic path planning and to multitask/multiagent RL, it cannot immediately be used online.

## 3 LEARNING CONTROL HIERARCHIES ONLINE

While there are many situations in which a hierarchical representation might plausibly be computed off-line, in a paradigm such as that of life-long learning, we would like our agents to be able to construct a hierarchical representation online. This representation would constitute an abstraction of the agent's experience *thus far*, and would need to be incrementally adjusted as the agent experiences more of the world and is better able to synthesis a global picture of their experience. The idea of incrementally adjusting an abstract representation of a task is highly intuitive. As people, we are often aware, not just of improving our performance on a task, but also of changing (and often simplifying) our thinking about the task as we gain more experience. Here, we will periodically increase the agent's representational capacity by incrementally adding subtask states. This procedure is similar to a number of methods for incremental learning on static problems in which additional hidden layers are added to a standard ANN architecture.

An important question that naturally arises is that of *when* we should incorporate a new subtask. Intuitively, since our subtasks constitute a spacial and temporal abstraction, we should add in a new subtask only when we have seen a sufficient number of new states. Suppose we wanted to construct an abstracted MLMDP with a state space $1/k$ the size of the base LMDP. We would then add in a subtask for every $k$ states the agent has seen. In practice we typically factor the state space such that $|S^0| = k^k$, which determines both the depth of the hierarchy (we will have $k$ layers) and the

abstraction factor for each layer (each layer has $1/k$ the number of state of the layer below it). We refer to this notion as the 'subtask genesis' condition. More generally, when the multi-scale structure of the domain is known *a priori*, the decomposition values for each layer may be explicitly specified. For example, in a rooms domain, if we know that each room contains 25 states, then we may choose a decomposition factor of $k^l = 25$ for the base layer.

## 3.1 LAYER-WISE UPDATES

Since we are incrementally adding states to our MDPs, we need a way of mapping the parameters from our old MDP onto our new MDP. Moreover, when a subtask is added, we need to make adjustments to both the current MDP (which has received a new subtask state) and the higher layer MDP (which has tacitly received a new interior state). Of course we will need to assume some default initialization for the new parameters here. As a guiding principle we seek to extend our MDPs in such a way as to assume as little as is possible about how the new state will integrate with the existing MDP. We do this by initializing the new state with small uniform default values, and allowing subsequent experience to shape the transition dynamics.

**Updating the current MDP**: the current MDP will now operate with a new subtask state $N_t^l \rightarrow N_t^l + 1$; depicted by the blue circles in Fig.(2). To define a consistent control task we are required to update the transition dynamics $P_t^l \in \mathbf{R}^{N_t^l+1}$ and the state costs $q_t^l \mathbf{R}$. Thereafter, the subtask component of the agent's task basis $Z_{:t}^l$, and controller $z_t^l$ will also need to be extended. We assume the previous components of $P_t$, governing the transition probabilities into the existing subtasks, are unchanged, and only those elements depicted by the blues lines need to be specified. We then initialize the transition probabilities into the new subtask to be uniform over the interior states. The new state cost is initialized to be the mean of the existing subtask costs, $q_t^l = 1/N_t^l \sum q_t(s_i)$. The new elements of the task basis and controller are initialized to one(s).

**Updating the higher layer MDP**: the higher layer MDP operates with a new interior state $N_i^{l+1} \rightarrow N_i^{l+1} + 1$, again depicted by the blue circle in Fig.(2). To define a consistent control task we are required to update the transition dynamics $P_i^{l+1}, P_b^{l+1}$, and the state costs $q_i^{l+1}$. Similarly, the interior component of the agent's task basis $Z_{:i}^{l+1}$, and controller $z_i^{l+1}$ will need to be updated. Again, we assume the previous components of $Pi, P_b$ are unchanged, and that only those elements depicted by the blue lines in Fig.(2) need to be specified. We assume that the new task cannot transition into a boundary state, initializing the new components of $P_b^{l+1}$ to zero. We also take the new components of $P_i^{l+1}$ to be uniform over the interior states. We initialize the state cost to the mean of the existing interior states $q_i^{l+1} = 1/N_i^{l+1} \sum q_i(s_i)$, and the new elements of the task basis and controller are initialized to one(s).

Essentially we have simply add a new state to our MDP, which is assumed to be loosely connected to all other states, and to have average reward. While this provides no immediate benefit, the additional representational capacity this provides to the higher layer will ultimately be utilized when these additional weights are updated.

## 3.2 EXPLORATION BOOSTS VIA POLICY COMPOSITION

Exploration with a hierarchical control architecture promises the possibility of dramatically improving exploration efficiency, since a 1-step transition in the higher layer MDP corresponds to a many-step transition in the lower layer MDP. In this way the agent is essentially diffusing via a heavy-tailed distribution; sometimes taking extended trajectories, and otherwise diffusing locally.

As discussed in Sec.(2.2), it is critical that the structure of the higher layer MDP is consistent with the lower layer MDP. In the off-line setting, this consistency requirement is achieved analytically via Eqns.(8,9) in Saxe et al. (2017). In an online setting, we need to make an initialization assumption about the transition structure. In practice it is important that we obtain a reasonable estimate for these dynamics otherwise the abstraction presented by the higher layer may be deleterious. Intuitively, operating with a poor estimate for the higher layer dynamics is equivalent to selecting, in the options framework, a poor options policy. This results in the agent having suboptimal local behaviour, until it transitions into an area in which the macro policy can be changed. In principle, we can always ensure that our estimate of the higher layer dynamics is sufficiently accurate by simply using a large enough

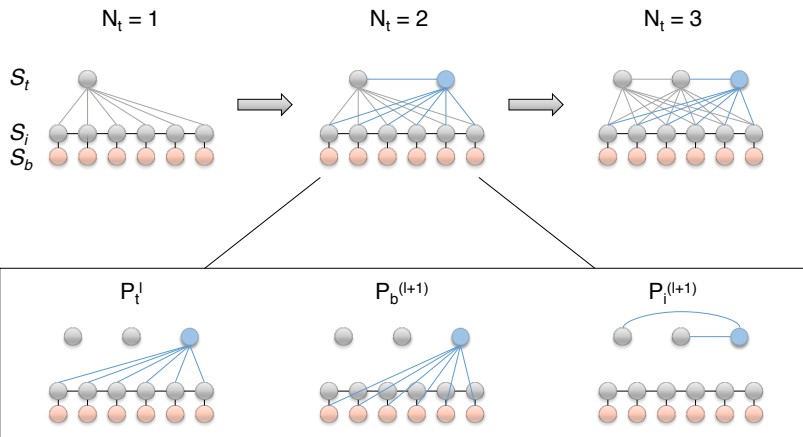

Figure 2: **Adding an additional subtask state**. The representational capacity of the hierarchy is incrementally increased by periodically adding a new subtask state. This state is suitably integrated into the problem, by specifying the associated transition and reward structures. Note that this procedure is defined layer-wise, and that additional subtask states are added at all levels of the hierarchy.

buffer. Of course, this in an intrinsically limited approached. We found that in practice it is possible to achieve good performance using a much smaller buffer size, if the agent is further equipped with a count based exploration boost. Intuitively this ensures that our agent explores enough of the surrounding states early on, to obtain a reasonable sample estimate of the higher layer dynamics.

Throughout the trajectory we maintain a vector of state visits $f$. A composite exploration policy $z_{exp}^l$ is constructed by computing $z_{exp}^l = Z^l w^l$, where $w^l = e^{f - \mu(f)}$ is the baseline adjusted state-frequency count. This simple construction makes rich use of the fact that policies compose in the MLMDP setting. Specifically, while the directed behaviours of a number of policies are aligned, the agent is strongly driven to pursue those initiatives. Alternatively, when directed behaviours conflict, the exploration boost is flat, and does not bias the agent's exploration. Said differently, the agent explores with consideration of their knowledge of the full space, not just their immediate surroundings. This approach over providing a count-based exploration boost over higher layer policies is conceptually similar to the count-based feature exploration considered in recent work Machado et al. (2018).

### 3.3 BALANCING DRIVERS: STRUCTURED AND UNSTRUCTURED EXPLORATION

A useful taxonomy for classical exploration strategies divides approaches into those that deliver undirected exploration (such as drawing states via the Boltzmann distribution), direct exploration which directly utilizes some of the agent's experience (such as state-count based methods), and model based methods. But combining the count-based exploration boost, the higher layer policies, and the base layer controller, we demonstrate that our agent utilizes elements of all three of these conceptual approaches concurrently.

At any point in time, our agent must balance multiple (potentially conflicting) drivers. Firstly, their actions may be driven by their primary policy $z_{base}$ which seeks to direct the agent to a goal state, but operates locally (missing the big picture). In the LMDP setting the optimal policy is stochastic and next-states are drawn from a Boltzmann distribution of the state-value function, scaled by the passive dynamics. Secondly, the agent's actions may be driven by a higher layer policy $z_{hl}$ which also seeks to direct the agent to a goal *region*, but operates more globally (missing finer details). The higher layer policy is delivered by an abstracted MLMDP which models the base problem. Finally, the agent may be driven by an exploration policy $z_{exp}$ which rewards infrequently visited states (at the cost of achieving the stated goal). This exploration boost is tantamount to a classical state-count based

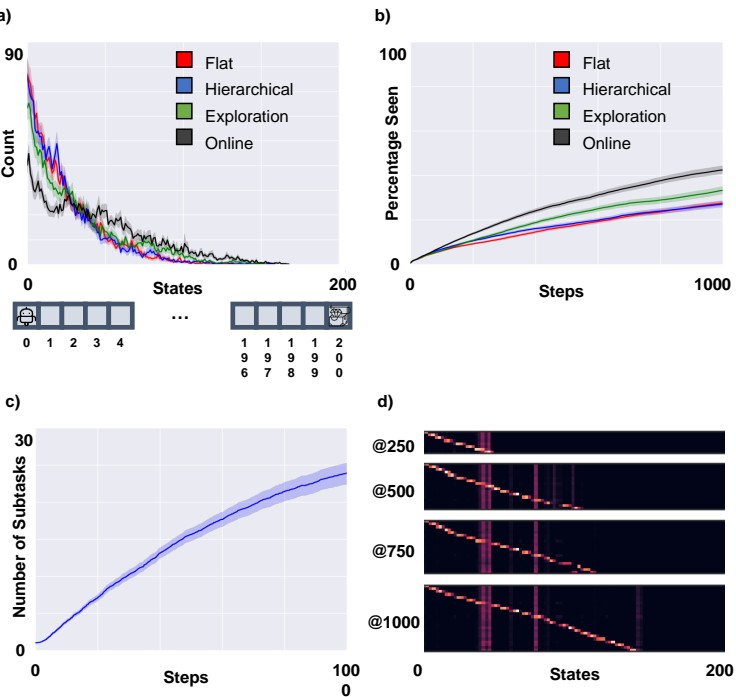

Figure 3: **Learning the hierarchy online can improve exploration.** a) The online policy is able to more reliably reach further than either the exploration policy, or the hierarhical policy (without exploration). b) With hierarchy, the agent is able to see more of the space earlier on. Rather than dithering at early states, the agent utilizes the higher layer policies to jump to the right boundary of its experience before exploring locally. c) The number of subtasks grows consistently with the number of steps taken by the agent. This suggests that the agent is being pushed to the frontier of its experience, seeing new states, and instantiating new subtasks. d) Heatmap plots of the exponentiated value function for the created subtasks, show how the subtasks tile the space. Notice that the number of subtasks grows as the agent takes more steps in the domain.

method. The compositionality of the MLMDP allows for these drivers to be employed concurrently in a natural way, by simply combining the relevant desirability functions. In practice our agent acts, at all times, under the composite policy:

$$z_{eff} = \alpha_1 z_{base} + \alpha_2 z_{exp} + \alpha_3 z_{hl}, \tag{2}$$

where the $\alpha_i$ are taken to be fixed weighting parameters. More generally, it is very likely that the agent would benefit from a dynamic update to this weighting scheme, although this is not considered in the present work. Intuitively the agent would benefit from a preference for an exploration policy early on, and for an abstracted policy later as it acquires more experience on the task.

## 4    RESULTS

There are two separate, but equally important results contained herein. The first is that the proposed method can uncover a deep hierarchical abstraction of the task ensemble in a completely online fashion. The abstractions we uncover with the online method converge to those uncovered analytically in Saxe et al. (2017), and in a batch-offline setting in Earle et al. (2018). Convergence here has a double meaning; the online method converges to the correct value estimates for the higher layer dynamics and reward structures as more samples are seen, but it also converges to the correct higher layer MLMDP structure by incrementally growing the abstracted MDP, adding states only as the

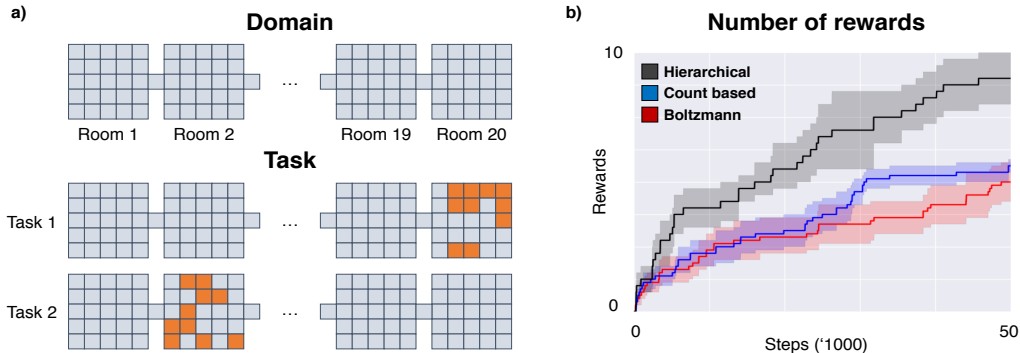

Figure 4: **Exploration with hierarchical policies** a) Our agent operates in a corridor-of-rooms domain. At any point in time, the task is specified as a single goal room, in which $50\%$ of the states are rewarded. When the agent reaches a rewarded state, the goal room is reset randomly. This corresponds to a foraging task in which the agent must explore both globally and locally. b) The number of goals reached by the agent in a finite horizon task of 50,000 steps. The hierarchical agent is better suited to the task owing to its ability to operate over an abstracted representation of the domain.

additional representation capacity is required. Once the hierarchy is in place, significant performance gains can be expected when the agent is faced with new tasks.

Secondly, we show that the process of learning the hierarchical structure online does not significantly slow down learning. In fact, it can actually improve exploration over the flat implementation! This is due to the fact that our method is incrementally growing and *solving* the higher layer abstractions online. In this way the agent is able to make use of a distilled form of its previous experience to efficiently traverse regions of the state space it has already explored. Mathematically, this has the effect of reducing the diffusion length between disparate regions of the state space.

In order to achieve the exploration improvements in the online setting, we further exploited the compositionality afforded by the MLMDP framework, to balance the directed behaviours of the base policy, higher layer policy, and exploration policy. The compounding effect of these drivers is disentangled in Fig.(3). Here, the agent operates in a 200-state 1D corridor. The agent begins always on the left-most end of the corridor, and explores the domain with a finite horizon of 1,000 steps. We measure the progress that the agent is able to make along the corridor under each exploration protocol. All results are averaged over 100 runs.

When the task is to reach a singleton goal state, the agent can do no better than explore each state individually. There is no implicit abstraction of the task ensemble or domain that aids learning. However this is typically not the case with real world tasks. A taxi driver seeks to drop off a passenger, and is rewarded for doing so; he is not concerned with the specific configuration of joint angles that resulted in the completion of the task. Said differently, there is a *region* in the parameter space in which rewards are clumped together corresponding to an abstract representation of the task. To mimic the property of tasks being represented by regions, rather than singleton states, we consider the following task setup; a simple scalable extension of the standard rooms domain. An agent operates in a corridor of 20 rooms, each of which is a 5-by-5 grid world. The rooms are connected to their left and right neighbours by a single state at the middle of the corresponding wall (see Fig.(4)). A goal room is then randomly selected and rewards are placed at $50\%$ of its states. The agent then explores the domain until it encounters one of the rewards. At this point the remaining rewards in the goal room are removed, and a new goal room is randomly selected. The agent continues in this vein for a finite time horizon of 50,000 steps. This task is motivated by a foraging metaphor in which an animal must explore a space both hierarchically (to find the right room) and locally (to find the right state). We compare the performance of our agent equipped with a learned hierarchy against two optimistically initialized Q-learning agents: the first drawing states from a Boltzmann distribution, and the second utilizing state-count based exploration boosts.

## 5 CONCLUSION

We present a method that is able to learn deep hierarchical control architectures in a fully online way. Moreover, we demonstrate in a toy domain that the agent is able to make use of its partial experience to-date to improve learning. In order to improve sample efficiency, we introduce a new exploration paradigm in multiple exploration initiatives operate concurrently to great effect. We compare the proposed method to some standard exploration strategies in a simple tabular domain requiring significant exploration. In future work we plan to extend the method to incorporate function approximators to extend the key conceptual advancements to large state and action spaces.

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
