# OpenReview forum: "Incremental Hierarchical Reinforcement Learning with Multitask LMDPs"
_ICLR.cc/2019/Conference_

### Official Review · AnonReviewer3 · 2018-11-02
**Nice idea and combination of methods. Difficult to assess significance**

**Rating:** 5
**Confidence:** 4

**Review:**

This work builds on the ICML paper from Saxe et al (2017) in which the compositionality property of LMDPs was exploited to solve multi-task hierarchies. The paper extends this work by proposing a method that learns incrementally such hierarchies instead of pre-defining them by design. Some experimental results illustrate the method on two toy problems, a 1D corridor and a corridor of rooms.

The paper deals with an interesting and hard problem. Learning hierarchies while solving an MDP is a much harder problem than solving the flat MDP or solving the hierarchical MDP. The authors leverage the compositionality of optimal controls of the LMDP framework to learn incrementally the hierarchies. Surprisingly, the proposed method not only learns those hierarchies, but also is more effective in terms of exploration.

On the positive side, the main idea is very interesting and has a lot of potential. The authors combine many techniques under the powerful framework of LMDPs such as hierarchical RL, low-rank factorization, and count-based exploration. The authors do a good job describing their approach (at a higher level).

On the negative side, the paper looks a bit incremental, given the prior existing work. I also found the paper unclear in many aspects lacking some relevant technical details (see below). The narrative is sometimes superficial or focused mainly on intuitions and analogies. Overall, it is difficult to assess the significance of this work and the results give the impression of limited applicability, beyond the described toy problem.

1- First, in order to combine optimal controls, the LMDPs need to be solved for each different boundary state, i.e., do you require to solve as many LMDPs as possible states? If that is the case, I don't think it makes sens to talk about exploration/exploration trade-off, since you really need to visit all states a priori.

2- I cannot understand what is learned and what is required a priori. the authors state that if "the multi-scale structure of the domain is known a priori, the decomposition (...) explicitly specified". What does exactly that mean? If what the method does is an incremental version of the low-rank factorization proposed in Earle at al (2018), I think the presentation can be better described in those terms.

2- Regarding exploration/exploitation tradeoff. From section 3, it seems that authors focus on a particular spatial problem and define already some exploration choices. But this means that the choice about when a state is integrated in the current MDP is already done, so no real trade-off exists?

3- The narrative in Section 3.2 is not very rigorous. The authors just mention the computational problem to keep consistency between layers and then just argue that "in practice" using count-based exploration everything works. I think a more principle approach is necessary.

4- Experiments I: what do the authors means by "exploration"? Is it just Boltzmann exploration? I can think of an exploration strategy that would choose an unseen state with probability 1 and would bring the agent to the goal in one shot.

5- Experiments II: I like the benchmark but, how does the result depend on the structure of the problem? What happens if I the rooms have very different sizes?

6- I miss some references that are very relevant to this work:

- the ICAPS paper "Hierarchical Linearly-Solvable Markov Decision Problems" by Jonson et al. seems to be the first proposing a hierarchical embedding of LMDPs.

- other factorization techniques exist, e.g., "Incremental Stochastic Factorization for Online Reinforcement Learning", Barreto et al (AAAI' 16), to uncover an MDP structure.

There are also some minor grammar mistakes:

"passive dynamics then become" -> "passive dynamics then becomes"
"reward function r" -> "reward function R"
"Howver" -> "However"
"the room contains" -> "the room that contains"
"have simply add" -> "have simply added"
"spacial and temporal" -> "spatial and temporal"
...

---

### Official Review · AnonReviewer2 · 2018-11-03
**This work extend previous work with multitask linear MDP for discovery of deep hierarchical abstractions from offline setting to a online setting. It lacks clarify in presentation and has limited experiments and anlaysis.**

**Rating:** 4
**Confidence:** 4

**Review:**

pros:
The experiment results demonstrate better performance of the proposed method.
Cons:
1. It does not compare with any existing HRL (with simple adaptation to multitask settings), except for Q learning under different exploration strategies.

2. Also the method is tested only on one type of domain-gridworld, which seems very limited, especially for supporting the claims, including “the proposed method can uncover a deep hierarchical abstraction of task ensemble in a complete online fashion” and “the process of learning the hierarchical structure online does not significantly slow learning”

3. There is no implementation details of the proposed in a algorithmic form.

4. There is also no ablation test or discussion of parameters that affect the performance of the experiment results, for example, learning rate, size of the domain.

5. There are a lot of notations used without definition. The paper should be more self-contained by providing accurate definition and citations.

For example, what are are Interior state,  boundary state, finite exit problem
Is V(s) computed with based on R_i or R, or both?
What is the control cost refer to? Is it related to R_i or R somehow?

6. Some references are cited wrongly.
Machado et al (2018) cited is not about HRL. Please double check
The references should be updated to reflect the latest information, for example,
It might be better to cite the AAAI17 version of the option-critic architectures
Between MDPs and Semi-MDP: A framework for Temporal Abstraction in Reinforcement learning should use the 1999 version


7. Minor issues:
task it is -> task is
Z^l \in R^{N\times N}
 Howver -> however

---

### Official Review · AnonReviewer1 · 2018-11-08
**Interesting ideas; contribution is marginal over previous work; lots of questions about hierarchy construction and utility**

**Rating:** 3
**Confidence:** 4

**Review:**

Major comments:

This paper builds on previous work in hierarchical LMDPs and extends the core ideas to an online setting.  Essentially, we incrementally construct a hierarchy by adding new states to upper-level MDPs every once in a while; these are loosely initialized and the parameters are then refined with additional experience.

Overall, I felt that this paper lacked a substantial enough contribution.

* The key contributions over previous work seems to be entirely contained in Sec. 3.1 and 3.2: (1) when we have visited k new states, add a new state to the hierarchy, and (2) initialize its parameters with intuitive values.

* To me, this level of contribution is below the bar for ICLR.  The ideas seem simplistic and likely to work only in the simplest of domains.

* The improvement over previous work is marginal.

* I think the paper is lacking in clarity.  I do not think I could re-implement the paper, given the level of detail presented.

* I was very disappointed in the experiments.  Not only were they on gridworld-like domains (see below), but it was not clear if the improvements were significant in any way.

* While I thought the discussion in Sec. 3.3 was interesting, it didn't seem to be a "contribution"; it felt like some adhoc thoughts.

* Keeping per-state counts is only workable in small state space domains.

Minor comments:

Please fix the formatting of your citations.

There are numerous typos and spelling errors.  Please correct them.

I am strongly opposed to the use of gridworlds, or anything like them, in modern RL research.  While many ideas work fine in small, toy domains, they simply do not scale.  As a field, we need to move past them and focus more on algorithms and ideas that have more practical relevance.

Pros:
+ Core ideas seem promising
+ Leveraging mathematical structure is a great strategy for constructing algorithms with desirable properties

Cons:
- Very limited experimental results
- Not clear if improvements are significant
- Hierarchy construction seems to be too limited to work in any reasonably sized problem
- No evidence, theoretical or otherwise, is given to suggest that this particular hierarchy construction method is any better than any other method

---

### Meta-Review · Area_Chair1 · 2018-12-14
**Interesting work, but contribution  seems incremental**

**Confidence:** 5
**Recommendation:** Reject

**Metareview:**

The paper studies an interesting problem with a reasonable solution.  However, reviewers feel that the technical contributions are somewhat incremental.  Furthermore, the empirical study would have been stronger with more proper baselines (simple adaptation to the multitask setting), and on problems beyond the simple grid worlds.  In addition, reviewers also find the presentation should be improved substantially.